# Position: Time to Close The Validation Gap in LLM Social Simulations

**Maximilian Puelma Touzel** [1]  **Sneheel Sarangi** [1 2]  **Aurélien Bück-Kaeffer** [1 2]  **Zachary Yang** [1 2]
**Jean-François Godbout** [1 3]  **Reihaneh Rabbany** [1 2]

## Abstract

LLM-based social simulations—in which many language model agents are situated in social situations and interact over multiple turns—are rapidly proliferating across policy analysis, epidemiology, and computational social science. Yet the field lacks consensus on how to validate these simulations, with evaluation methods that are few, underdeveloped, fragmented, and rarely shared across disciplines. We argue this creates a serious risk: premature deployment of unvalidated simulators in high-stakes domains. Our position is that the field must pivot from expansion to consolidation, prioritizing methodological standardization—shared benchmarks, open data, and reproducible evaluation protocols grounded in social science and complex systems research. We outline a concrete research program organized around specific learning problems/benchmarks, providing a path toward answering the fundamental question: when are LLM social simulations useful modelling objects?

## 1. Introduction

Large Language Models (LLMs) constitute a potential step forward in simulating human social systems, modelling human behaviour at a degree of social complexity and grounding that exceeds existing approaches, covering phenomena central to culture and society. A motivation to pursue such approaches comes from viewing LLMs as cultural technologies, providing access to large amounts of cultural information (Farrell et al., 2025). For such phenomena, they would supplant both traditional agent-based modeling, where minimal agent complexity limits expressivity, and deep reinforcement learning simulations, where a blank-slate burden limits

grounding to real social phenomena (Lake et al., 2017).

This technology promises to produce world models (Ding et al., 2025) (such as social world models), useful for understanding mechanisms, testing counter-factuals, and making predictions. Among distinct objectives, the *simulationist* perspective narrows the aim to reproducing phenomena with the highest possible fidelity (Vezhnevets et al., 2025). Researchers across disciplines are now developing LLM-based simulation approaches spanning generic social world modelling, social media, public health, and policy analysis. We survey this literature in Section 2; it is scattered across disciplines, bringing heterogeneous goals and norms around quality and rigor.

A natural consequence of this discordant development is slow convergence towards consensus about what these models can do. Seemingly conflicting results on core questions—whether LLMs reproduce survey responses, whether agent groups produce genuine emergent behaviour—often trace back to divergent methodology and problem framing rather than substantive disagreement. In parallel, evaluation-centric approaches from social science and feature-centric approaches from software engineering have developed largely in isolation. Both are necessary, neither alone sufficient. Assessing the real utility of LLM-based social simulations requires their careful integration within a unified methodological framework. The natural vehicle is machine learning methodology: well-defined learning problems supported by shared data, open code, and standardized benchmarks.

**Position:** Validation of LLM-based social simulations has not kept pace with their proliferation. The field must pivot from expansion to consolidation—adopting shared benchmarks, open data, and reproducible evaluation grounded in social science—before these tools can responsibly inform high-stakes decisions.

Summarizing our paper's contributions, we:

1. Provide a *modelling language* to ground evaluation of a class of models we term Silicon Societies.
2. Express a list of *evaluation fallacies* in the literature and phrase them in our language.
3. Outline a *frictionless reproducibility* approach and pro-

---

[1]Mila - Quebec Artificial Intelligence Institute, Montréal, Canada [2]McGill University, Montréal, Canada [3]Université de Montréal, Montréal, Canada. Correspondence to: Maximilian Puelma Touzel <puelmatm@mila.quebec>.

*Proceedings of the $43^{rd}$ International Conference on Machine Learning*, Seoul, South Korea. PMLR 306, 2026. Copyright 2026 by the author(s).

vide an example simulator project illustrating best practise.

4. Layout a set of *learning problems* inspired by a set of target properties and evaluations designed to measure them (stability, polarization, narratives).

5. Engage with *alternative views* and sketch a *call-to-action* about how researchers from different disciplines can participate in our proposed structured research community.

## 2. Related Work

Research on LLM-based social simulation has expanded rapidly but unevenly. We organize the relevant literature around four recurring themes: claims and counter-claims about emergent phenomena, the construction of social media simulation platforms, the use of LLMs as persona stand-ins, and an emerging critical literature on validation. Each raises a tension between ambitious capability claims and the methodological infrastructure needed to substantiate them.

**Emergent Phenomena in LLM Societies.** A first strand asks whether populations of LLM agents can produce behaviour that resembles human social dynamics without being explicitly programmed to do so. The seminal demonstration of generative agents (Park et al., 2022) suggested LLM agents in a shared environment could exhibit coordinated, lifelike behaviour, and recent work scales this to argue for human-like polarization (Piao et al., 2025a) and other macro-level patterns. Such claims face methodological challenges. Barrie & Törnberg (2025) show that several behaviours reported as "emergent" are observationally indistinguishable from training-data leakage; independent replication efforts report substantial failures to reproduce headline results (Curvo et al., 2025); and Song et al. (2025) document systematic behavioural degradation under sustained multi-agent pressure. The shared message is that "emergence" is doing heavy lifting without an agreed operationalisation, and the same observation is often consistent with very different generative mechanisms.

**Social Media Simulation Platforms.** A second strand builds general-purpose simulators, most prominently for social media environments where machine-readable interaction traces are abundant. Platforms such as OASIS (Yang et al., 2024), the systems of Piao et al. (2025a), and the simulator of Ng & Carley (2025) provide infrastructure for instantiating large populations of LLM agents at scale, complemented by domain-specific deployments in public health (Shi et al., 2026; Chopra et al., 2024) and computational social science (Kozlowski & Evans, 2025; Puelma Touzel et al., 2025). A persistent gap is that these simulators are rarely tested against well-established empirical results about online behaviour. Robertson et al. (2024) document how so-cial media systematically distorts users' perception of social norms—an effect with substantive policy implications that any credible simulator of online discourse should reproduce, yet that is almost never used as a validation target. Generic simulators are thus often advertised as broadly applicable while leaving domain-specific validation to downstream adopters.

**LLMs as Persona Stand-ins.** A third strand investigates the use of LLMs to simulate individual human respondents—the atomic unit of any silicon society. Persona-conditioned generation has been explored for survey response and behavioural prediction (Li et al., 2025a; Abdurahman et al., 2025), and Kozlowski & Evans (2025) discuss the promise and peril of LLMs as stand-ins for human subjects. The empirical record is mixed. Systematic biases in LLM-generated personas have been documented along demographic and ideological dimensions (Schröder et al., 2025; Namazova et al., 2025), and a growing literature emphasises structural limits: LLM agents lack embodied experience, temporal continuity across episodes, and the material incentives that shape human social behaviour (Binz et al., 2025; Bück-Kaeffer et al., 2025). These limitations matter because they bear directly on which simulation tasks can plausibly be validated, and they help explain why the literature on whether LLMs can reproduce survey responses appears to conflict: results depend strongly on the elicitation protocol, the subgroup considered, and the measurement instrument.

**Focus on Validation.** A fourth, more recent strand explicitly foregrounds validation. Larooij & Törnberg (2025) argue that much of the silicon-society literature fails to evidence *operational validity*—the property that a simulator reproduces, rather than merely resembles, the target phenomenon. Wallach et al. (2025) frame evaluation of generative AI systems as a measurement problem, importing the conceptual toolkit of psychometrics and survey methodology, and Cui et al. (2025) make a closely related argument from the social-science side. Madden (2025) translate these arguments into practical evaluation guidelines. A complementary line of work attempts to characterise validity through compact principle sets such as PIMMUR (Zhou et al., 2025b); while valuable as shared vocabulary, such principle sets are not yet paired with the shared benchmarks, datasets, and reproducible protocols that have driven progress in adjacent areas of machine learning.

Across these four themes the picture is one of rapid capability claims, scattered validation, and a field that has not agreed on what a successful silicon society would look like. Our position is that closing this gap requires treating silicon societies as objects of machine-learning methodology: well-defined learning problems supported by shared data,

open code, and standardised benchmarks, grounded in the measurement traditions of social science. The remainder of the paper develops this position, beginning with a minimal formal abstraction in Section 3.

# 3. Silicon Societies

The four themes above share a common object of study but lack a common vocabulary. Before articulating evaluation fallacies, learning problems, or reproducibility standards, we need a minimal abstraction sharp enough to state what a silicon society is and is not.

We use *Silicon Societies* to refer to simulations of human social interactions with a large amount of implicit (i.e. latent) cultural context selectively made explicit using LLMs. Two features of this definition do real work. First, the cultural context is *latent in the model rather than specified by the modeller*: this distinguishes silicon societies from traditional agent-based models, where every social fact must be hand-coded, and from blank-slate reinforcement learning populations, where none is available at initialisation. Second, only a *selected* subset of that context is made explicit at any point in the simulation, through prompts, scaffolding, or environment design. The modeller's craft lies in choosing that subset for the phenomenon under study.

This framing makes the scope of our position precise. Task-oriented multi-agent AI systems—coding, web, and computer-use agents—do not model the human environment into which their outputs deploy and do not seek to reproduce human behaviour; they are out of scope, though some of their evaluation methods remain relevant (Vijayvargiya et al., 2025). In scope are systems whose target phenomena are shaped by social reasoning over implicit and explicit norms, or by other facets of social intelligence that emerge among groups of agents. Emergent social intelligence is thus a central topic, and design within this scope is governed by what Vezhnevets et al. (2025) call the *simulationist* objective: empirical fidelity over a specified subset of social phenomena, with microscopic detail required only insofar as the phenomena of interest demand it.

A practical consequence of the simulationist objective is that machine-readable digital environments occupy a privileged position. They supply large quantities of human interaction data against which simulator behaviour can be grounded and validated. Social media is the canonical example, and—as the platforms surveyed in Section 2 illustrate—it is also where the gap between simulator construction and simulator validation is currently widest.

We now introduce a minimal formal abstraction capturing what silicon societies have in common. More general formulations exist (Ferrarotti et al., 2026), but ours is specific enough to state important validation problems precisely, and provides the language used throughout the paper.

## 3.1. Formal Abstraction

In a high-level description, a silicon society induces a probability distribution over system trajectories. A trajectory records the evolution of the environment and agent interactions over time, including what agents observe and how they act. Representing simulation dynamics as statistics over trajectories is useful in the non-stationary settings these complex simulations inhabit, allowing effects to be traced to causes and even mechanisms to be identified, at least coarse-grained ones (Cranmer et al., 2020).

Formally, let

$$\tau = (s_0, \{o_0^i, z_0^i, a_0^i\}_{i=1}^n, s_1, \ldots, s_T) \,,$$

denote a $n$-agent system trajectory of length $T$, where $s_t$ represents the environment state at time $t$, and $o_t^i$, $z_t^i$, and $a_t^i$ denote the observation, internal state, and action of agent $i$.

A *silicon society* defines a distribution,

$$p(\tau \mid \mathcal{X}, \boldsymbol{\Theta}) \,,$$

where $\mathcal{X}$ denotes the simulation specification (e.g., prompt templates, orchestration logic, environment rules), and $\boldsymbol{\Theta}$ denotes both agent and environment parameters (*e.g.*, LLM weights, temperatures, and persona/environment initializations).

This abstraction is intentionally minimal. It hides architectural details and treats the simulator as a generative process over trajectories. Our critique applies to any system that induces such a distribution, regardless of whether agents are implemented via prompting, fine-tuning, tool use, explicit planning modules, or hybrid approaches.

Evaluations of silicon societies are then functions of trajectories,

$$\mathcal{L}(\tau),$$

which may assess properties such as behavioral realism, stability over time, coordination strength, or alignment with empirical data. Learning or tuning a simulator—whether by adjusting prompts, model parameters, or environment design—can be understood as optimizing expected evaluation outcomes under this distribution.

Many claims made in the literature, including claims about emergence, coordination, or norm formation, are ultimately statements about statistical properties or dependencies within $p(\tau \mid \mathcal{X}, \boldsymbol{\Theta})$. Making this objective explicit allows such claims to be stated precisely and, in principle, tested.

## 3.2. A Representative Simulator Decomposition

We now describe one common instantiation of the abstraction above, using a machine learning–oriented formalism

inspired by stochastic games and prior work on LLM-based agent simulations (e.g., Concordia (Vezhnevets et al., 2023)). This decomposition is representative rather than normative; many variants are possible.

A multi-agent simulation consists of $n$ agents interacting within a shared environment. Each agent is implemented using a generative model (e.g., an LLM) that samples outputs conditioned on inputs and parameters. Given an input $x$ (such as a prompt with context) and parameters $\theta$, the model generates

$$y \sim p(\cdot \mid x, \theta),$$

where $y$ can be text, structured data, or an action specification.

**System Components**  At time $t$:

- $s_t$ – environment state (locations, objects, social context)
- $o_t^i$ – agent $i$'s observation of the environment: $o_t^i = \mathcal{O}^i(s_t)$
- $a_t^i$ – agent $i$'s action (speech, movement, posts). Together, these actions form the joint action vector, $\boldsymbol{a}_t$.
- $\theta^i$ – LLM parameters for agent $i$ (model type, weights, temperature, static+initial persona context)
- $\mathcal{A}^i(\cdot)$ – agent $i$'s action generation function (prompt structure)
- $\theta_{\text{Env}}$ – environment parameters (model type, rules, static+initial environment context)
- $\mathcal{T}(\cdot)$ – environment state update function (prompt structure or rules)
- $\boldsymbol{\Theta} = (\theta_{\text{Env}}, \theta^1, \ldots, \theta^n)$ – all agent+environment parameters.

In addition to having implicit internal states (LLM activations), agent models can have explicit (e.g., natural language) internal states:

- $z_t^i$ – agent $i$'s internal state (beliefs, memories, goals, plans), and
- $\mathcal{Z}^i(\cdot)$ – agent $i$'s internal state update function (prompt structure).

**Dynamics**  Each timestep involves agent computation following a perception-cognition-action cycle along with environment computation providing observations and resolving action effects:

1. **Think:** Update internal state based on observation and prior state

$$z_{t+1}^i \sim p(\cdot \mid \mathcal{Z}^i(o_{t+1}^i, z_t^i), \theta^i), \qquad (1)$$

where $\mathcal{Z}^i(o_{t+1}^i, z_t^i)$ constructs the prompt, *e.g. "Given {interaction} & {retrieved memories/beliefs}, what would {name} think?"*.

2. **Act:** Generate action based on internal state

$$a_{t+1}^i \sim p(\cdot \mid \mathcal{A}^i(z_{t+1}^i), \theta^i), \qquad (2)$$

where $\mathcal{A}^i(z_{t+1}^i)$ constructs the prompt, *e.g. "Given {interaction} & {thoughts}, what would {name} do?"*.

3. **Environment Update:** An environment simulator is used to update the world based on all actions

$$s_{t+1} \sim p(\cdot \mid \mathcal{T}(s_t, \boldsymbol{a}_t), \theta_{\text{Env}}), \qquad (3)$$

where $\mathcal{T}(s_t, \boldsymbol{a}_t)$ encodes update rules and/or constructs the prompt for LLM-based updating, *e.g. "Given {the state}, and {names} attempted {actions}, what happens?"*.

Simplified variants may omit explicit internal state and generate actions directly from observations and history, trading interpretability for computational efficiency.

Under this decomposition, the simulator induces the trajectory distribution introduced in Section 3.1. Different choices of prompts, parameters, or environment rules correspond to different distributions over trajectories and, consequently, different simulated social dynamics.

The benefit of this abstraction is not technical novelty but analytical clarity. Making the generative object explicit allows claims about silicon societies to be precisely formulated and evaluated, and clarifies what it means to validate, compare, or transfer simulations across settings—questions that motivate the critiques and fallacies in the next section.

## 4. Common Evaluation Pitfalls in Silicon Societies

LLM-based agent simulations incur orders-of-magnitude higher computational costs than classical agent-based models (ABMs) (Samsi et al., 2023). To justify this expense, they must demonstrate clear advantages in explanatory power or empirical fidelity. However, recent *Silicon Society* literature has been accussed of "failing to adequately evidence operational validity"(Larooij & Törnberg, 2025). Simulators are often engineered at scale (thousands of agents (Yang et al., 2024)), yet remain hard to use for scientific inference because their alignment with real-world behavior is weakly evidenced or unknown.

Below, we group common pitfalls into four salient axes. More general machine learning issues (e.g., cherry-picking / contamination) appear in Table 1 but are not expanded upon, being non-specific to this setting. Most of these pitfalls reflect methodological immaturity, rather than possible fundamental issues with the approach. The latter questions, *e.g.* how much LLMs capture the latent structure of social interactions, are still open.

| Axis | Fallacy | Framework Component | Core Issue |
|------|---------|---------------------|------------|
| I. Distributional Validity | Face Validity | $p(\tau \mid \boldsymbol{\Theta}) \approx p^*(\tau)$ | Low-dimensional plausibility $\neq$ full trajectory distribution |
| | Aggregate Validation | $\mathbb{E}[\psi(\tau)]$ vs. $p(\tau)$ | Matching moments $\neq$ matching distributions |
| | Cherry-Picking[†] | $p(E \mid \boldsymbol{\Theta})$ for event $E$ | Post-hoc selection of favorable events |
| II. Agent–Human Correspondence | Human Proxy Assumption | $p(a \mid s)$ | LLM behavior assumed human without evidence |
| | Social Desirability Bias | $p(a \mid \mathcal{A}, \theta)$ | RLHF skews action distributions |
| | Demographic Representation | $\mathcal{A}^i(d^i)$ | Prompting demographics $\neq$ embodied agents |
| III. Design Justification | Untested Design Choices | $\mathcal{X}, \boldsymbol{\Theta}$ | Complexity added without ablation or justification |
| | Prompt Sensitivity | $\mathcal{Z}^i, \mathcal{A}^i$ | Semantic invariance violated under rephrasing |
| | Black-Box Gap | $z_t^i$ | Internal reasoning unverifiable or uninterpretable |
| | Reproducibility Failure | $(\mathcal{Z}, \mathcal{A}, \mathcal{T}, \boldsymbol{\Theta}, \ldots)$ | Key components underspecified or unstable |
| IV. Emergence & Diversity | Homogeneity | $\mathrm{Var}[f(\tau)]$ | Shared $\theta$ collapses agent diversity |
| | Single-Model Dependence[†] | $\boldsymbol{\Theta}$ fixed | Model-specific artifacts mistaken for phenomena |
| | Circular Evaluation | $\theta^{\text{judge}} \approx \theta$ | Shared biases inflate evaluation scores |
| | Sycophancy | $p(a \mid \mathcal{A}, \theta, r)$ | Researcher intent leaks into behavior |
| Generic ML Pitfalls[†] | Contamination | $\tau^* \in D_{\text{train}}$ | Memorization mistaken for generalization |
| | Temporal Validity | $T_{\text{cutoff}} \in \theta$ | Knowledge-time mismatch |
| | Causal Confounding | $p(U \mid s_t, T, \theta)$ | Unconfoundedness assumptions violated |

*Table 1.* Common evaluation fallacies in Silicon Society simulations, grouped by axis and mapped to components of the simulator formalism. [†]Denotes pitfalls not specific to multi-agent LLM simulations, but included for completeness.

### Axis I: Distributional Validity

A pervasive pitfall is equating qualitative plausibility with distributional correctness. Many works rely on anecdotal examples, hand-picked trajectories, or compelling visualizations as evidence of success, implicitly assuming that $p(\tau \mid \mathcal{X}, \boldsymbol{\Theta}) \approx p^*(\tau)$, where $p^*(\tau)$ denotes the (unknown) real-world process (Larooij & Törnberg, 2025; Zhou et al., 2025a).

In practice, evaluations often focus on low-dimensional projections (denoted $\psi$), such as individual conversations, summary statistics, or emergent narratives, rather than trajectory distribution itself. However, agreement on moments or marginal statistics does not imply agreement on the full distribution. This leads to *aggregate validation* fallacies, where matching $\mathbb{E}[\psi(\tau)]$ is mistaken for matching $p(\tau)$.

Of course there are meaningful notions of inexact matching, *e.g.* distinguishability by humans or a family of classifiers (Pagan et al., 2025), but such studies are rare. Matching issues are exacerbated by vague problem definitions "simulating society" or "modeling social media" (Yang et al., 2024; Piao et al., 2025b; Park et al., 2023; Surve et al., 2023; Vezhnevets et al., 2023; Kaiya et al., 2023; Park et al., 2022) and ad-hoc metrics that prevent cross-paper comparison. In our formalism, this corresponds to validating isolated samples of $\tau$ rather than defining and optimizing a principled evaluation function $\mathcal{L}(\tau)$.

### Axis II: Agent-Human Correspondence

Many Silicon Society projects implicitly assume that LLM-based agents are reasonable proxies for human cognition and behavior. Yet empirical evidence paints a mixed picture. Studies comparing LLMs and humans across behavior (Bück-Kaeffer et al., 2025; Ma, 2025), neural representations (Holton et al., 2025; Pinier et al., 2025; Fedzechkina et al., 2025; Aw et al., 2024; Holton et al., 2026; Zhou et al., 2025c; Kwon et al., 2025), and psychological constructs (Schröder et al., 2025) yield inconsistent results.

Despite this, several works proceed as though the correspondence holds by default (Abdurahman et al., 2025; Piao et al., 2025b; Wang et al., 2023). In our notation, this amounts to assuming $p(a_i|s) = p_{\text{sampled human}}(a|s)$ without direct empirical support. Attempts to mitigate these gaps, either through alignment techniques or cognitive modeling, remain limited in adoption and are themselves contested (Binz et al., 2025; Chen et al., 2025b; Kong et al., 2026; Schröder et al., 2025; Namazova et al., 2025). Even the weaker assertion that $p(a_i|s) \sim p_{\text{sampled human}}(a|s)$ on the phenomena of interest is rare.

Without explicitly testing agent–human correspondence, downstream conclusions about social dynamics rest on an unstable foundation. This has strong implications when investigating how LLMs would behave in the real world (Ren et al., 2026; Orlando et al., 2025) using simulated environments.

**Axis III: Unjustified Simulator Design Choices**

A distinct but closely related pitfall concerns simulator design itself. Silicon societies typically consist of many interacting components (e.g., persona initialization, memory systems, planning modules, environment, and orchestration logic), each adding complexity to $\mathcal{X}$ and $\Theta$.

A common fallacy is to treat this complexity as self-justifying. Components are often motivated by cognitive theories or narrative plausibility (even in the seminal paper (Park et al., 2023)), but whose necessity is never checked, e.g. through ablation studies. Although recent large-scale simulators show ablations are nearly always feasible (Yang et al., 2024; Piao et al., 2025b), such analysis remains rare.

This lack of validation leads to two complementary pathologies. Some works introduce elaborate mechanisms to address problems that are assumed, rather than demonstrated, to exist; others implicitly assume these mechanisms are unnecessary, again without testing the assumption. In both cases, $\mathcal{X}$ and $\Theta$ are expanded without evidence of improvement under any evaluation function $\mathcal{L}(\tau)$, obscuring causal attribution and hindering principled comparison across simulators.

Concrete examples illustrate this. Vezhnevets et al. (2023) and Wang et al. (2023) propose cognitively inspired agent frameworks, the former grounding design choices in cognitive psychology. While theoretically motivated, such justification is insufficient absent evidence that these components improve task-relevant or societally meaningful metrics. Conversely, Yang et al. (2024) adopts a minimal, memory-based design without ablation-based justification. To our knowledge, whether human-inspired cognitive components yield measurable improvements in simulation quality remains an open empirical question.

**Axis IV: Emergence, Coupling, and Diversity**

A defining motivation for Silicon Societies is the study of emergent social phenomena. However, many simulations inadvertently suppress emergence through architectural choices that induce excessive homogeneity. Shared model weights, similar prompts, and strong RLHF priors reduce variance across agents (Wu et al., 2025b; Jiang et al., 2025), leading to collective behavior that reflects model bias rather than interaction-driven dynamics.

In formal terms, emergence requires violations of conditional independence: $p(\boldsymbol{a}_t \mid s_t) \neq \prod_{i=1}^{n} p(a_t^i \mid s_t)$. Yet few works test this explicitly. While techniques to increase diversity exist (Nguyen et al., 2025), and its importance has been argued for (Robertson et al., 2024), empirical demonstrations linking diversity to improved simulation fidelity remain scarce and have been contested (Barrie & Törnberg, 2025).

**Towards Proper Validation**

Recent work has begun to argue that persona simulation itself requires a scientific methodology (Li et al., 2025a). While these efforts focus primarily on individual agents, introducing environments and multi-agent interaction induces additional complexities and emergent effects (Orlando et al., 2025; Curvo et al., 2025; Schroeder et al., 2026) that demand additional, distinct validation strategies (Song et al., 2025).

Existing surveys and methodological critiques largely agree that current practices are insufficient (Larooij & Törnberg, 2025), and recent proposals offer some practical guardrails (Madden, 2025). What remains lacking is adoption and iterative improvement. Establishing shared standards for specifying simulators, justifying design choices, and evaluating trajectory-level behavior is essential if Silicon Societies are to mature into a reliable scientific tool rather than an exercise in plausible storytelling.

## 5. The Solution is Frictionless Reproducibility

Machine learning has grown at an unprecedented pace. What explains this? Platt asked the same of molecular biology fifty years ago, pointing to rigorous hypothesis-based testing (Platt, 1964). But this "strong inference" approach is weaker for complex systems—particularly social ones—where causes cannot be isolated to individual components (Anderson, 1972). Donoho offers a more compelling answer for machine learning's rapid progress: "frictionless reproducibility", an interconnected set of research practices and incentives that make methodology explicit and efficiently leverage community consensus mechanisms (Donoho, 2023; Recht, 2024). In this section, we contribute elements of this practise for the case of silicon society research.

### 5.1. Learning Problems

**Agent Models** Agent models are a primary target of machine learning. Here we list some non-mutually exclusive agent training paradigms:

1. *Model Training*: Parameters can be fine-tuned from a base model $\theta^i = \theta_{\text{base}} + \Delta\theta^i$, using techniques such as Supervised Fine Tuning (SFT) or Reinforcement Learning (RL).

2. *Model Steering*: Steering vectors $\Delta x^i$ can be used to bias model activations $x^i$, controlling the value of the output logits of the model. This is used to structure the internal model representation (Chen et al., 2025a).
3. *Prompt Engineering*: The structure of prompt functions can be trained, *e.g.*, by
   - structure learning a component network ($z$ now being a vector of natural language statements and $\mathcal{Z}$ now updating all components with conditional dependence given by a component graph), and
   - including data directly in the prompt, e.g. $\{a_j^i\}_{j=1}^D$ for $D$ in-context learning samples.
4. *Persona Learning*: Creating realistic agents may require grounding on real-world data. Personas can be condensed from existing human data and provided to the model via the above methods.

## 5.2. Core Learning Objectives

We now describe several learning problems shared across silicon societies, ordered from most concrete to most structural.

**Next-Action Prediction** The most direct and operationally convenient learning problem is next-action prediction. Given the environment state and interaction history, the agent predicts its next action:

$$\mathcal{L}_{NAP} = -\log p(a_{t+1}|s_t, a_t, \ldots, s_1, a_1)$$

This framing collapses "thinking" and "acting" into a single observable step, aligning well with real-world datasets in which only actions and environment updates are logged. Social media platforms are a canonical example: users observe a post and may reply, react, repost, or ignore.

NAP Evaluation. Metrics can be defined at multiple granularities:

1. *Individual level*: action classification (e.g., F1-score), semantic similarity of generated content (cosine similarity of embeddings, Jaccard similarity over high-IDF terms), or divergence measures (e.g., Jensen–Shannon).
2. *Population level*: distributional alignment using aggregated statistics, kernel density estimates, or Monte Carlo rollouts compared against historical data (e.g., KL or JS divergence).

Projects such as BluePrint (Bück-Kaeffer et al., 2025) instantiate this paradigm explicitly, emphasizing population-level fidelity rather than per-agent optimality. Matching here should cover both the content and action type (Gatta et al., 2026).

While next-action prediction is easy to train and evaluate, it is agnostic to internal reasoning structure and offers limited guarantees about long-horizon dynamics.

**Component Structure Learning** Beyond action prediction, some systems aim to learn or impose structure over agent internal state $z_t^i$. We distinguish two broad approaches:

1. Explicit structure learning, where internal state is represented in natural language or symbolic components (e.g., beliefs, goals, memories), possibly with learned dependency graphs and update rules.
2. Implicit structure learning, where internal structure is encoded in parameters (fine-tuned weights, steering vectors) without easily-interpretable state variables.

These approaches correspond to different inductive biases:

1. persona inference via bottom-up emergence or top-down role imposition,
2. static versus dynamically updated internal state, and
3. interpretable but brittle representations versus opaque but flexible ones.

Objectives in this inquiry regime are often indirect, such as narrative alignment, behavioral consistency, or constraint satisfaction, rather than likelihood maximization.

**Environment Initialization and Orchestration** Learning problems also arise at the level of the environment:

1. **Initialization**: choosing initial states $s_0$ (e.g., network structure, topic distribution, agent personas) that reproduce realistic downstream behavior.
2. **Interaction Orchestration**: determining whether interactions are narration-driven, dialogue-driven, or event-driven, and how information propagates across agents.

Although these components are often treated as fixed design choices, they implicitly define strong priors over trajectories and can dominate downstream outcomes. One such design choice is whether agents update simultaneously or sequentially and in what order.

## 5.3. Stability and Long-Horizon Behavior

Silicon societies differ most from persona research (centering around one-shot settings, such as survey responses) in that they admit interaction dynamics among agents. This temporally-extended simulation setting raises the question of long-horizon behavior (Wu et al., 2025a; Song et al., 2025). Even when local objectives (e.g., next-action prediction accuracy) are satisfied, small modeling errors may compound over time, leading to distributional drift or unrealistic equilibria.

This framing enables several important distinctions:

1. Local fidelity vs. global stability: a simulator may match empirical action distributions at short horizons while diverging over longer rollouts.

2. Training vs. deployment mismatch: objectives optimized under short trajectories may not control long-term population behavior.

3. Evaluation beyond snapshots: stability metrics depend on trajectories, not single-step predictions.

While most existing silicon societies do not explicitly optimize, let alone evaluate stability, making stability assumptions explicit clarifies what claims can—and cannot—be supported by current evaluation practices. While model complexity precludes the kind of mathematical control in dynamical systems research, the concepts of ergodic theory do provide a language to describe and classify long-term behaviour.

### 5.4. Simulator Properties

What does a frictionless reproducible silicon society simulator code project look like? At its core, it should be structured for training against research-driven evaluation objectives, with highly interoperable components. Based on our formulation, such a simulator's configuration should have 5 main components:

- simulator (engine logging and execution)
- model (genAI models, model parameters, $\Theta$)
- environment , $\mathcal{T}, \theta_{\text{Env}}$ (interaction rules)
- scenario (agent models $\mathcal{A}_i, \mathcal{Z}_i$, shared knowledge)
- evaluation $\mathcal{L}$ (metrics, probes, statistical operations)

Each component shold be configurable via schema-constrained plain text, with variants easily created through default overrides. The simulator should output human-readable action logs for evaluation. Probe systems survey agents longitudinally, feeding responses to the evaluation function $\mathcal{L}$. Scenario generation should be straightforward with step-by-step instructions. The project should follow open science practices: version-controlled with strict formatting and testing requirements, readable contribution guidelines, and emerging best practices for coding agents. Project management should be largely automated, letting researchers focus on model development and evaluation rather than tooling. We suggest allocating a fixed compute fraction to evaluation, prioritizing lightweight longitudinal probes for early-stage development, and reserving expensive distributional tests for mature studies. Such code projects are appearing, *e.g.* the Silicon Society Sandbox (SiliSocS[1]) simulator (Sarangi et al., 2026).

## 6. Alternative Views

Our position is constructive: we not only identify the dilemma but propose a timely solution. Alternative views to our position can take a few forms:

**Arguments Against Benchmarking.** A common objection is that benchmarks are inherently gameable. This concern is valid, but incentive structures are improving rapidly. For example, PeerBench (Cheng et al., 2025) combines sealed execution, heldout splits, item banking with rolling renewal, and delayed transparency, and operates alongside open/dynamic benchmarks to reduce strategic overfitting (Kiela et al., 2021).

A deeper critique is that ML benchmarks are ill-suited to large, complex social systems, where ground truth is often ambiguous or unavailable. It is unclear, for instance, how to objectively rank a simulated response to a policy intervention or global event. From this perspective, benchmarking risks imposing a false precision that misrepresents the epistemic status of social science. We agree that naïve ranking is inappropriate. However, evaluation need not reduce to scalar scores; it can characterize model behavior, surface sensitivities, and clarify which assumptions drive outcomes. Social science already operates productively without definitive ground truth, relying on comparative analysis and typologies; the framework we propose explicitly draws on these traditions.

**The Real Problem is Deployment, not Methodology.** Another view holds that the primary risk is not the absence of benchmarks, but the uncritical deployment of these simulators by decision-makers. The remedy, on this account, is governance and stakeholder education rather than additional ML infrastructure. We agree that standards alone cannot prevent misuse. However, shared evaluation norms are themselves communicative: they make limitations legible, create common reference points for critique, and signal appropriate use to downstream stakeholders.

**Ethical and Technical Challenges of Obtaining Ground Truth.** A key motivation for *Silicon Societies* is studying scenarios too costly, impractical, or unethical to examine with human subjects. Here relevant datasets may be sparse, sensitive, or unavailable, making validation against real-world ground truth infeasible. We acknowledge this limitation, but caution that deploying unvalidated models in high-stakes domains is itself problematic. Complex phenomena like emergence can, we think, we suitably captured: violations of conditional independence across agents (Axis IV), and approaches even for smallish systems (Daniels et al., 2017) that suggest concrete operationalizations tied to our trajectory formalism. In cases lacking ground truth access, researchers can pursue proxy-based validation calibrating on historical trajectories with held-out periods or cross-domain transfer as partial validation, plus synthetic ground truth from well-understood ABMs as a controlled testbed. When grounding is impossible, this should be stated explicitly and the strength of resulting claims attenuated ac-

---

[1]https://github.com/sandbox-social/silisocs

cordingly. Where feasible, ethical data frameworks should be developed and preferred (Bück-Kaeffer et al., 2025).

**Validation is Premature and May Stifle Development.** A related argument is that the field is too young to know which assumptions matter, and that early validation risks locking in the wrong abstractions. We agree that exploratory work is essential. Nonetheless, some degree of evaluation is necessary even at early stages to ensure progress is cumulative rather than idiosyncratic. Curvo et al. (2025) and Barrie & Törnberg (2025) both demonstrate how poor evaluation wastes effort and stalls progress. Focusing on evaluation can in fact catalyze progress, especially when evaluating over many dimensions. Provisional and lightweight evaluation frameworks can evolve alongside the field without foreclosing future methodological shifts.

**Are Social Simulations a Good Use Case for LLMs?** Given the cost and complexity of LLMs, it is important to clarify when LLM-based social simulations are well motivated. Traditional agent-based models capture diffusion processes such as social contagion and epidemics (Guilbeault et al., 2017; Sambaturu et al., 2020), but struggle with phenomena rich in social meaning. Deep reinforcement learning agents exhibit general coordination capabilities (Stooke et al., 2021), yet remain far from human social complexity. One-shot or few-turn LLM persona studies probe implicit social knowledge (Li et al., 2025a), and are informative when interaction and dynamics are not central.

We argue that LLM-based social simulations are a truly novel and useful approach to modelling, and are especially valuable for phenomena characterized by:

- *Collective network effects* emerging from many interacting agents;
- *Social norms and sanctioning* shaping behavior (e.g., family planning, public health compliance, political participation);
- A central role for *language*, including semantic abstraction and linguistic ambiguity.

## 7. Call to Action

Realizing this vision requires coordinated action. ML researchers should adopt validation frameworks, report sensitivity and variance metrics, and build on community standards rather than ad hoc procedures. Social scientists can identify suitable benchmark datasets, advise on domain-appropriate evaluation criteria, and challenge oversimplified claims about simulation capabilities. Venues (NeurIPS, ICML, AAMAS) should require validation protocols for acceptance and create tracks for methodological work. Funding agencies can include validation requirements in proposals and prioritize interdisciplinary teams and method-

ological rigor. Industry labs should adopt validation benchmarks internally, contribute to open infrastructure, and be transparent about model limitations for social simulation. To catalyze this transition, we propose forming an interdisciplinary working group to deliver three foundational resources:

1. A benchmark suite spanning survey response prediction, economic games, opinion dynamics, and crisis behavior, with held-out test sets and contamination controls;
2. An open-source evaluation toolkit implementing standardized metrics for distributional validity, variance calibration, and sensitivity analysis; and
3. An author checklist and reviewer guidelines for LLM social simulation papers.

These resources would provide the shared infrastructure necessary to transform a fragmented literature into a cumulative research program capable of answering the fundamental question: when, if ever, can LLM social simulations be trusted?

## 8. Responsible Research

This research program centers around advancing LLM simulations. Like all powerful technologies, LLMs are dual-use. An accurate social simulator could be misused for manipulation campaigns, persuasive advertising, engineered misinformation, or content exploiting social media algorithms. Early work on design iteration using these technologies suggests such applications may be credible (Li et al., 2025b; Duetting et al., 2025). Simulations might also train manipulator agents or generate fake engagement.

These concerns align with existing literature on LLM social risks. Researchers have already used LLM simulations to study AI-coordinated influence campaigns (Orlando et al., 2025), build influential social media bots (Jin & Guo, 2025), and model adversarial bot dynamics (Le et al., 2022). Evidence shows LLMs are highly persuasive, even regarding elections (Lin et al., 2025) or conspiracy theories (Costello et al., 2026), while LLM-generated disinformation—a documented threat to democracies (McKay & Tenove, 2020; Badawy et al., 2018; Givi et al., 2024)—was already difficult to detect in 2023 (Chen & Shu, 2024). These concrete risks add to the AI safety risks of multi-agent AI systems (Hammond et al., 2025).

We share these concerns. However, malicious actors freely test tactics on real people, while safety researchers face ethical constraints. These simulators would provide safe testing environments for developing defenses without harming anyone. Pursued responsibly, this tool could level the playing field, though risks should be measured and limited.

## Acknowledgements

The authors acknowledge funding support from the Future of Life Institute and IVADO. The authors sincerely thank the three anonymous reviewers for helping improve the paper. We are grateful for their constructive critiques. The authors thank the members of the Complex Data Lab for helpful discussion.

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

# A. Full Trajectory Factorization

For completeness, we provide the explicit factorization of the trajectory distribution induced by the simulator described in Section 3.2.

Let $\mathcal{X}$ denote the set of prompt constructors and orchestration logic, and let initial states be drawn from $p(s_0, z_0^1, \ldots z_0^n)$. The joint distribution over a trajectory $\tau$ of length $T$ factorizes as:

$$p(\tau \mid \mathcal{X}, \boldsymbol{\Theta}) = p(s_0, z_0^1, \ldots z_0^n) \prod_{t=0}^{T-1} \Bigg[ \underbrace{p(s_{t+1} \mid \mathcal{T}(s_t, \boldsymbol{a}_t), \theta_{\text{Env}})}_{\text{environment}} \times$$

$$\prod_{i=1}^{n} \underbrace{p(o_t^i \mid s_t)}_{\text{observe}} \cdot \underbrace{p(z_{t+1}^i \mid \mathcal{Z}^i(o_{t+1}^i, z_t^i), \theta^i)}_{\text{think}} \cdot \underbrace{p(a_{t+1}^i \mid \mathcal{A}^i(z_{t+1}^i), \theta^i)}_{\text{act}} \Bigg]$$

This factorization makes explicit the conditional dependencies between environment dynamics, agent observations, internal state updates, and action generation. While not required for the conceptual arguments in the main text, it enables precise reasoning about independence assumptions, intervention effects, and evaluation metrics.

# B. Simulation Visualization

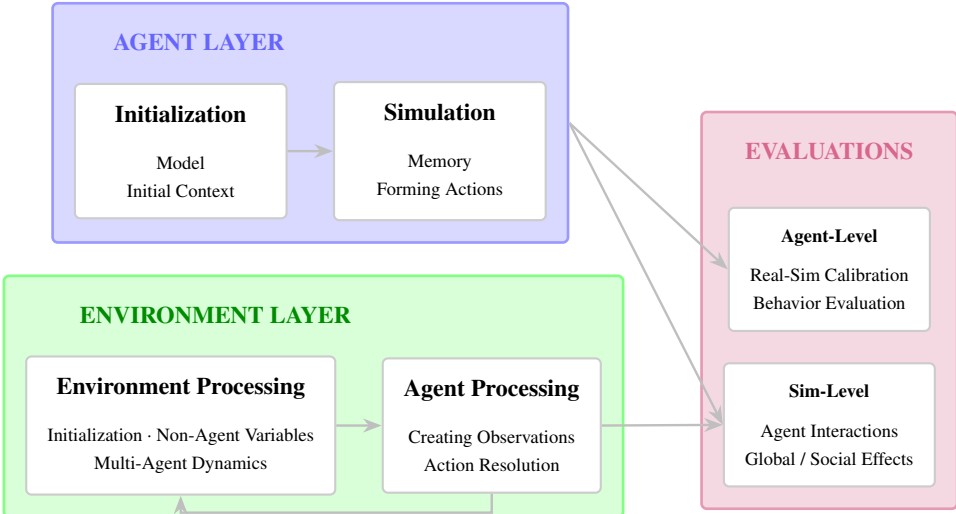

*Figure 1.* A visualization of the components involved in the simulation task. Arrows depict flow of information/computation.

