# OpenReview forum: "Position: Time to Close The Validation Gap in LLM Social Simulations"
_ICML.cc/2026/Position_Paper_Track — ICML 2026 Position Paper Track regular_

### Official Review · Reviewer_gNgZ · 2026-02-25

**Significance:** 3
**Argument Clarity:** 3
**Ethics Flag:** Yes
**Rating:** 5
**Confidence:** 4

**Questions:**

Can you please refer to weakness

**Alternative Views Section:**

Yes

**Compliance With Llm Reviewing Policy A Conservative:**

Affirmed.

**Discussion Potential:**

3

**Ethical Review Concerns:**

the topic on social simulation may requires ethical review (flag so nothing are missed)

**Ethics Review Area:**

["Discrimination / Bias / Fairness Concerns", "Responsible Research Practice (e.g., IRB, documentation, research ethics)"]

**Final Justification:**

The response addressed my main concerns, particularly by clarifying the scope of the argument and sharpening the distinction between correctable methodological failures and more fundamental open questions. While I still think some aspects could be developed further in the final version, I now view the paper as a good and timely position contribution, and I am increasing my score to 5.

**Paper Summary:**

This position paper argues that current LLM-based (multi)-agent “social simulations” lack rigorous validation and risk being treated as scientific models without sufficient empirical grounding. The authors formalize such systems as probabilistic trajectory generators and identify common evaluation fallacies (e.g., narrative plausibility, unjustified complexity, and false emergence). They advocate for standardized benchmarks, reproducibility, and long-horizon validation to close what they term the “validation gap.” The formulation of "Silicon Societies" is interesting.

**Position:**

Yes

**Position In Title:**

Yes

**Related Work:**

3

**Strengths And Weaknesses:**

**Strengths**

1. Timely and important topic. LLM multi-agent simulations are rapidly growing, and methodological scrutiny is necessary.

2. Clear articulation of risks. The paper systematically categorizes evaluation failures (e.g., distribution validity fallacy, fake emergence). (although I think there are many ways of catergorise these)

3. Constructive framing. Rather than rejecting the field, the authors propose concrete learning tasks (next-action prediction, structure learning, environment learning) as a path forward.

**Weaknesses**

1. Assumption of scientific modeling objective: The critique presumes that LLM social simulations are intended as faithful models of real societies. However, many works treat them as exploratory or generative tools rather than inferential instruments.  (in ML, the meaning of model can sometimes be the problem that we care about, so maybe the models are not meant to be faithful anyway?) The paper would benefit from clarifying when strict validation is required versus when heuristic utility is sufficient.

2. Benchmark feasibility: Social systems often lack well-defined ground truth. The proposal for standardized evaluation is compelling, but more discussion is needed on how benchmarks avoid oversimplifying inherently open-ended phenomena. (any maybe some suggestions on how / what can be achieved will be useful)

3. Scope of critique: The paper risks conflating weak empirical practice with conceptual invalidity. It would be useful to distinguish between immature methodology and fundamental impossibility.

**Support:**

2

---

> ### Author Rebuttal · Authors · 2026-03-31
>
> We thank Reviewer gNgZ for their thoughtful review and recognition of our constructive framing. We address each weakness below.
>
> ## W1: Assumption of scientific modeling objective
>
> We agree this distinction needs to be sharper. Our position is scoped to applications where fidelity matters: policy analysis, epidemiological modeling, computational social science, and any setting where outputs inform claims about human behavior. Without minimal validation, findings risk being artifacts of architecture or prompt design — a concern empirically grounded by Barrie & Törnberg (2025), who show apparent emergence can be indistinguishable from data leakage.
>
> We acknowledge that many legitimate uses fall outside our scope: generating compelling narratives (Vezhnevets et al., 2025), studying LLM agents as objects in their own right (Hammond et al., 2025), game design, and stress-testing AI systems (Orlando et al., 2025; Schroeder et al., 2026). We will revise the manuscript to explicitly distinguish the simulationist objective (where our framework applies) from these other uses.
>
> ## W2: Benchmark feasibility and oversimplification risk
>
> We share this concern. Our mitigation is twofold:
>
> **Scenario diversity:** We advocate a portfolio of evaluation scenarios across qualitatively different domains (political deliberation, marketplace behavior, public health, online discourse) — not a single benchmark.
>
> **Evaluation taxonomy:** A multi-dimensional validity profile (behavioral realism, temporal stability, agent diversity, design sensitivity) rather than a scalar score, following established methodology (Sargent, 2013) and the PIMMUR principles (Zhou et al., 2025).
>
> We also acknowledge genuine boundary conditions: phenomena depending on non-textual exchange or social knowledge poorly represented in training data pose real limits. We will state these boundaries explicitly, noting that identifying where the approach reaches its limits is itself a valuable outcome of systematic evaluation.
>
> ## W3: Distinguishing immature methodology from fundamental impossibility
>
> Our position: most pitfalls in Section 3 reflect methodological immaturity, not fundamental impossibility.
>
> **Evidence for methodological origin:** Agent homogeneity arises from shared weights and RLHF priors — design choices, not theoretical limits. Existing techniques already address this: min-p sampling (Nguyen et al., 2025), persona vectors (Chen et al., 2025a; Fedzechkina et al., 2025), and fine-tuning on heterogeneous data (Buck-Kaeffer et al., 2025). Similarly, the absence of ablations and reliance on face validity are correctable. That 89% of studies in the Zhou et al. meta-study failed at least one validation principle suggests substantial room for improvement.
>
> **Areas where fundamental questions remain open:** Two stand out. First, social construction under underspecification — agents may construct aberrant structures reflecting training data regularities rather than coherent social logic. Whether this is addressable through better grounding remains empirically open. Second, causal identification — even perfect trajectory matching does not guarantee correct mechanism identification when the state space is compressed (Cranmer et al., 2020).
>
> We will revise to explicitly separate addressable methodological failures from genuinely open questions. This distinction strengthens our position: precisely because most failures are methodological, the field has substantial room for improvement — but only with proper evaluation infrastructure.
>
> ## References
>
> Cranmer, K., Brehmer, J., & Louppe, G. (2020). The frontier of simulation-based inference. *Proceedings of the National Academy of Sciences*, 117(48), 30055–30062.
>
> Sargent, R. G. (2013). Verification and validation of simulation models. *Journal of Simulation*, 7(1), 12–24.
>
> Zhou, J., Huang, J., Zhou, X., Lam, M. H., Wang, X., Zhu, H., Wang, W., & Sap, M. (2025). The PIMMUR Principles: Ensuring Validity in Collective Behavior of LLM Societies. *arXiv preprint arXiv:2509.18052*.

---

> > ### Author Rebuttal · Reviewer_gNgZ · 2026-03-31
> >
> > Thank you for the rebuttal. The response addressed my main concerns in a substantive way, particularly by clarifying the scope of the argument and sharpening the distinction between correctable methodological failures and more fundamental open questions. While I still think some aspects could be developed further in the final version, I now view the paper as a solid and timely position contribution, and I am updating my score to 5.

---

### Official Review · Reviewer_PCaq · 2026-03-02

**Significance:** 4
**Argument Clarity:** 2
**Rating:** 3
**Confidence:** 4

**Questions:**

N/A

**Alternative Views Section:**

Yes

**Compliance With Llm Reviewing Policy A Conservative:**

Affirmed.

**Discussion Potential:**

2

**Final Justification:**

The rebuttal has addressed my main concerns. I changed my evaluation to a higher score.

**Paper Summary:**

This paper introduces the position that validation practices for LLM-based social simulations have not kept pace with their rapid proliferation. The authors formalize “Silicon Societies” as generative processes over social interaction trajectories and identify common evaluation fallacies. They analyze common evaluation pitfalls in silicon societies from four perspective, and conclude this paper with alternative views, call to action and conlusion.

**Position:**

Yes

**Position In Title:**

Yes

**Related Work:**

1

**Strengths And Weaknesses:**

Strength

(1) Interesting and timely topic

(2) Well-organized

Weaknesses

(1) I feel the related works are not adequately discussed. I would suggest add some discussions on the main stream social simulation topics and how they may differ from real society evolutions.

(2) Evaluations may restrict simulations. The human society has almost unlimited topics to simulate, yet building evaluation frameworks may limit the scope of social simulation. And, following works may only focus on the simulations where benchmarks have been build. In this sense, the evaluations actually slowed down the development of simulations. This paper should discuss on this.

(3) Holistic and fair evaluations are always hard to build. This paper should discuss some potential obstacles and problems simulator developers may meet, and provide some potential solutions.

(4) This paper did not mention anything on the cost of evaluation, as social simulation and evaluations may cost too much tokens  for accurate evaluation.

**Support:**

2

---

> ### Author Rebuttal · Authors · 2026-03-31
>
> We thank Reviewer PCaq for recognizing the timeliness and organization of our paper. We respond to each weakness below.
>
> ## W1: Related works not adequately discussed
>
> We will restructure related work into four thematic clusters:
>
> - **Emergent phenomena:** Park et al. (2023); Piao et al. (2025a) on emergent coordination; Barrie & Törnberg (2025) showing claimed emergence is indistinguishable from data leakage; Curvo et al. (2025) on replication failures; Song et al. (2025) on behavioral degradation under multi-agent pressure.
> - **Social media modeling:** Yang et al. (2024); Piao et al. (2025b); Ng & Carley (2025) on simulation platforms; Robertson et al. (2024) on how social media distorts norm perception — an effect simulators rarely validate against.
> - **Persona consistency:** Li et al. (2025a); Abdurahman et al. (2025) on persona simulation; Schroder et al. (2025) and Namazova et al. (2025) on systematic biases in LLM-generated personas; Kozlowski & Evans (2025) on the promise and peril of LLM stand-ins.
> - **Validation criticism:** Larooij & Törnberg (2025) on failure to evidence operational validity; Wallach et al. (2025) framing evaluation as a measurement challenge; Madden (2025) on practical evaluation guidelines.
>
> We will also discuss how LLM agents lack embodied experience, temporal continuity, and material incentives (Binz et al., 2025; Buck-Kaeffer et al., 2025), and cover attempts to characterize validity through principle sets (e.g., PIMMUR; Zhou et al., 2025).
>
> ## W2: Evaluations may restrict simulations and slow development
>
> This aligns with the "Validation is premature" view in Section 5, which we will expand. We agree exploration is valuable, but it becomes counterproductive when important findings are indistinguishable from unreliable ones. Barrie & Törnberg (2025) show reported emergent behaviors may be memorization artifacts, and Curvo et al. (2025) demonstrate replication failures in published cooperation dynamics. Without evaluation, the field cannot distinguish signal from noise.
>
> ML's own history shows benchmarks *accelerate* progress: ImageNet (Russakovsky et al., 2015), GLUE (Wang et al., 2019), and shared evaluation infrastructure made progress legible and cumulative. Donoho (2023) argues "frictionless reproducibility" drives ML's rapid iteration. Our example simulator in Section 4.4 demonstrates that modern tooling makes for easy systematization.
>
> ## W3: Holistic evaluations are hard to build
>
> Our mitigation (shared concern with Reviewer gNgZ) is twofold:
>
> - **Scenario diversity:** No single benchmark should dominate. Our example simulator includes multiple scenario types with instructions for generating new ones, preventing overfitting to any single domain.
> - **Evaluation taxonomy:** Rather than a scalar score, we advocate a multi-dimensional taxonomy (behavioral realism, temporal stability, agent diversity, design sensitivity), following established methodology (Sargent, 2013; Zhou et al., 2025).
>
> We will also address practical obstacles: ground-truth scarcity for counterfactual scenarios, evaluating emergent properties, and benchmark gaming — where mechanisms like PeerBench's sealed execution and rolling item renewal (Cheng et al., 2025) offer promising mitigations.
>
> ## W4: Cost of evaluation not discussed
>
> Costs are significant for long rollouts, combinatorial sensitivity analysis, and large agent populations. We will discuss this in revision.
>
> However, the cost of *not* evaluating must be weighed against this. An unvalidated simulation informing policy or public health decisions carries substantial downstream risk. If one has the budget to run a simulation but not to evaluate it, one has spent resources generating data of unknown reliability — a poor allocation by any standard. Evaluation must be factored into simulation sizing from the outset, as is standard in computational science (Oberkampf & Roy, 2010). Furthermore, inference costs are declining roughly an order of magnitude per year at equivalent capability (Samsi et al., 2023), making today's cost concerns increasingly transient.
>
> ## References
>
> Oberkampf, W. L., & Roy, C. J. (2010). *Verification and Validation in Scientific Computing*. Cambridge University Press.
>
> Russakovsky, O., et al. (2015). ImageNet Large Scale Visual Recognition Challenge. *International Journal of Computer Vision*, 115(3), 211–252.
>
> Sargent, R. G. (2013). Verification and validation of simulation models. *Journal of Simulation*, 7(1), 12–24.
>
> Wang, A., et al. (2019). GLUE: A Multi-Task Benchmark and Analysis Platform for Natural Language Understanding. *ICLR*.

---

> > ### Author Rebuttal · Reviewer_PCaq · 2026-04-01
> >
> > Thanks for the clarifications. I'm clear on W1.
> >
> > On W2, what I'm concerning is your evaluation may limit development to try their best to get SOTA on benchmarks rather than exploring new topics. ImageNet played an important role in computer vision, but also made people spend decades only on image classification. The evaluation sort of "guided" further development.
> >
> > W3 can you elaborate some pending modifications/drafts on the next version?
> >
> > W4 I think the authors addressed this concern. Thanks

---

### Official Review · Reviewer_nHW4 · 2026-03-13

**Significance:** 3
**Argument Clarity:** 3
**Rating:** 5
**Confidence:** 4

**Questions:**

Please refer to the weakness part.

**Alternative Views Section:**

Yes

**Compliance With Llm Reviewing Policy A Conservative:**

Affirmed.

**Discussion Potential:**

3

**Final Justification:**

The rebuttal is comprehensive. My questions have been addressed. My positive score remains unchanged.

**Paper Summary:**

This position paper argues that LLM-based social simulation (“Silicon Societies”) is advancing rapidly in scale and application while lagging significantly in validation methodology, creating a “validation gap” that risks misleading scientific conclusions and downstream decision-making. The authors advocate for a shift from expansion to methodological consolidation, proposing that the field should prioritize standardized evaluation frameworks, shared benchmarks, open simulation specifications, and reproducible protocols.

The paper introduces a minimal formalization that treats social simulators as generators of trajectory distributions, enabling clearer reasoning about what claims (e.g., emergence, alignment, realism) actually mean in statistical terms. Using this abstraction, the authors identify several recurring evaluation pitfalls in current literature, including reliance on face validity instead of distributional validity, untested assumptions about agent–human correspondence, lack of ablations for architectural design choices, and failure to assess long-horizon stability and interaction-induced effects.

To address these issues, the paper proposes adopting “frictionless reproducibility” as a methodological goal: simulation systems should be modular, shareable, and easy to replicate, allowing controlled comparisons, sensitivity analysis, and cumulative progress across studies. The authors further suggest reframing validation as a set of machine-learning-style learning problems and benchmarks (e.g., next-action prediction, structural inference, long-horizon stability evaluation), and outline a reference architecture for reproducible simulation infrastructure.

**Position:**

Yes

**Position In Title:**

Yes

**Related Work:**

3

**Strengths And Weaknesses:**

Strength
1. Addresses a clear and timely pain point.
The paper tackles a genuine and increasingly visible issue in the LLM-based social simulation community: the absence of authoritative benchmarks, unified evaluation standards, and broadly accepted validation protocols. The authors argued that the field currently suffers from a “validation gap,” where rapid system-building and narrative-driven demonstrations outpace rigorous methodological grounding. This concern is highly relevant to ICML, especially as LLM-based multi-agent systems, simulation environments, and societal-scale modeling are gaining traction. By explicitly articulating this gap, the paper provides a focal point for a conversation that is already emerging informally within the community. In that sense, the topic is both important and likely to stimulate discussion among ML researchers working on multi-agent systems, LLM evaluation, and simulation-based decision tools.

2. Clear and coherent position.
The paper states a clear position: the field should shift from expansion-driven innovation toward methodological consolidation, emphasizing shared evaluation infrastructure, reproducibility, and principled validation. The stance is consistently developed throughout the manuscript and does not drift into unrelated commentary.

3. Useful formal abstraction (trajectory view).
The abstraction of LLM-based social simulators as generators of trajectory (rollout) distributions is a strength. It provides a minimal yet flexible formal lens through which claims about emergence, realism, or alignment can be clarified as statements about properties of induced distributions. While not technically novel, this abstraction improves conceptual clarity and strengthens the internal coherence of the argument.

4. “Frictionless Reproducibility” as a concept.
The introduction of “frictionless reproducibility” is a notable contribution of the paper. Rather than treating reproducibility as a general ideal, the authors argue for building infrastructure that makes modular ablations, controlled component replacement, stability and sensitivity analysis, and cross-paper comparison low-cost and routine. This shifts reproducibility from an aspirational principle to a practical, operational goal embedded in research workflows.

Weakness
1. Limited empirical evidence supporting some claims.
While the reasoning is generally sound, several central claims (e.g., long-horizon instability, systematic overreliance on face validity, widespread architectural overfitting) are supported more by theoretical argument and anecdotal observation than by systematic empirical review. For example: (1) The claim that long-horizon rollout instability is a serious and underexamined issue is plausible, but concrete documented failures or comparative studies are not extensively discussed; (2) The discussion of evaluation pitfalls would benefit from a more structured survey of representative works demonstrating these patterns.

2. Scope boundaries could be sharper.
The paper focuses on LLM-based interactive social simulations, but the boundary between social simulation as trajectory rollout, static generative modeling, and causal or mechanistic social modeling is not always sharply delineated. Because the proposed framework centers on trajectory distributions, it may under-specify how it applies to mechanism-oriented modeling, structural or equilibrium-based simulations, and non-interactive or partially observed systems. Clarifying the limits of applicability would strengthen the position and help situate the framework more precisely within the broader landscape of social and computational modeling approaches.

**Support:**

3

---

> ### Author Rebuttal · Authors · 2026-03-31
>
> We thank Reviewer nHW4 for their careful and constructive review. We are glad the reviewer recognizes the timeliness of our position, the utility of the trajectory abstraction, and the operationalization of frictionless reproducibility.
>
> ## W1: Limited empirical evidence supporting some claims
>
> We agree. We’ll strengthen the empirical grounding in revision by these additional examples and by reorganizing related work into thematic sections.
>
> **Long-horizon instability:** Wu et al. (2025) demonstrate distributional drift and behavioral collapse over extended rollouts. Curvo et al. (2025) show that previously reported cooperation dynamics fail to replicate under tighter experimental controls. Song et al. (2025) find that agent behavior degrades under sustained multi-agent social pressure.
>
> **Overreliance on face validity:** Larooij & Törnberg (2025) systematically review generative social simulations and conclude the field broadly fails to evidence operational validity. Barrie & Törnberg (2025) show that reported "emergent" behaviors are observationally indistinguishable from data leakage — directly undermining face-validity-based evaluation. Wallach et al. (2025) and Madden (2025) independently call for more rigorous standards. Zhou et al. (2025) do a validation meta-study finding 89% of works fail at least 1 of a set of 6 simulation validation principles they propose, some of which are related to face validity.
>
> **Architectural overfitting:** Yang et al. (2024) demonstrate that ablations are feasible even at scale, yet most simulators (Park et al., 2023; Wang et al., 2023; Piao et al., 2025b) introduce cognitively motivated components without controlled ablation against task-relevant metrics — precisely the pattern our Axis III targets.
>
> We will make these contrasts more structured and explicit in the revision.
>
> ## W2: Scope boundaries could be sharper
>
> We appreciate this observation. Our focus on trajectory distributions was deliberate: they are the most fundamental representation of simulator output, and in principle sufficient to recover any higher-level quantity. This generality is what makes them useful as a common formal language. However, we acknowledge that the mapping to researchers' actual objects of interest is not always straightforward, and we will clarify the boundaries in Section 2.1.
>
> Two cases merit explicit treatment:
>
> **Stationary behavior.** When researchers care about convergence to (even a macroscopic) steady-state distributions, statistics over states may be more parsimonious than full trajectory analysis. For one-shot behaviour (e.g. surveys), this is tractable. For multi-turn settings, however, non-stationarity in the form of long transients arising from the complex full or simplified coarse-grained representations means a turn-indexed trajectory is then needed.
>
> **Causal inference.** Trajectory matching is a strong constraint but does not alone identify causal mechanisms. A compressed observable (e.g., a majority-opinion flip) may be consistent with multiple generative processes — threshold effects, influence cascades, or network topology. This is a general challenge for complex simulators with intractable likelihoods, as articulated in simulation-based inference (Cranmer et al., 2020). More efficient surrogate simulators (e.g. of coarse-grained variables) can be trained on trajectory data from the full multi-agent simulator. Mechanisms are more interpretable in low-dimensional, coarse-grained representations and defining the feasible space of mechanisms more tractable, potentially enabling causal inference at a macroscopic level. On the other hand, sufficiently granular trajectories offer high discriminating detail to tease apart the origin of observed behaviour. So, yes, trajectory matching must be complemented by ablations, counterfactual interventions, and sensitivity analysis to support mechanistic claims.
>
> We will revise Section 2.1 to state explicitly where the trajectory lens applies most naturally and where complementary representations are preferred.
>
> ## References
>
> Cranmer, K., Brehmer, J., & Louppe, G. (2020). The frontier of simulation-based inference. *Proceedings of the National Academy of Sciences*, 117(48), 30055–30062.
>
> Zhou, J., Huang, J., Zhou, X., Lam, M. H., Wang, X., Zhu, H., Wang, W., & Sap, M. (2025). The PIMMUR Principles: Ensuring Validity in Collective Behavior of LLM Societies. *arXiv preprint arXiv:2509.18052*.

---

> > ### Author Rebuttal · Reviewer_nHW4 · 2026-04-04
> >
> > Thank you for addressing my questions and concerns. I will stick to my positive score.

---

### Review · Ethics_Reviewer_zd8G · 2026-03-27

**Recommendation:** No remediation action needed

**Ethics Issue:**

This work was flagged for an ethics review with the following justification: "the topic on social simulation may requires ethical review (flag so nothing are missed)" with the categories "Discrimination / Bias / Fairness Concerns, Responsible Research Practice (e.g., IRB, documentation, research ethics)".

In my view, this work introduces no specific ethics issues beyond those inherent to the topic (social simulation). The work does already allocate significant space to discussing ethical considerations and potential harms (see sections 5 and 7). The discussion and modeling of such implications are also entangled with the core position that the work makes regarding the validation gap in evaluations of LLMs used for social simulations. As a point of constructive feedback, the authors could consider revision to emphasize discussion of potential harms as they related to discrimination, bias, or fairness concerns, but in my view this is not so severe a limitation so as to require remediation as a part of the ethics review process. Furthermore, the flag regarding responsible research practice is not particularly relevant to this work given that it is a position paper that does not include any experiments or use any data.

---

### Decision · Program_Chairs · 2026-04-30

**Decision:**

Accept (regular)

**Comment:**

This paper addresses a very timely topic (the use of LLMs as social simulators) with a clear position and call to action. The paper could be strengthened with improvements to the alternative views discussed with reviewers during the rebuttal period. Presentation of this position at ICML could promote very important interdisciplinary discussions.